# DEEP IDEATION: DESIGNING LLM AGENTS TO GENERATE NOVEL RESEARCH IDEAS ON SCIENTIFIC CONCEPT NETWORK

## ABSTRACT

Novel research ideas play a critical role in advancing scientific inquiries. Recent advancements in Large Language Models (LLMs) have demonstrated their potential to generate novel research ideas by leveraging large-scale scientific literature. However, previous work in research ideation has primarily relied on simplistic methods, such as keyword co-occurrence or semantic similarity. These approaches focus on identifying statistical associations in the literature but overlook the complex, contextual relationships between scientific concepts, which are essential to effectively leverage knowledge embedded in human literature. For instance, papers that simultaneously mention "keyword A" and "keyword B" often present research ideas that integrate both concepts. Additionally, some LLM-driven methods propose and iteratively enhance research ideas using the model's vast internal knowledge, but they fail to effectively leverage the valuable scientific concept network, limiting the grounding of these ideas in established research. To address these challenges, we propose the **Deep Ideation** framework, which integrates a scientific network that not only captures keyword co-occurrence but also incorporates contextual relationships between keywords, providing a richer scientific foundation for LLM-driven ideation. Our framework introduces an explore-expand-evolve workflow for Deep-Ideation which integrates several key components to iteratively refine research ideas. Throughout this workflow, we maintain an Idea Stack to track research progress across iterations. To guide this search and evolution process, we integrate a critic engine trained on real-world reviewer feedback, providing continuous signals on the novelty and feasibility of generated ideas. Experimental results across multiple AI domains show that our approach significantly improves the overall quality of generated ideas by **10.67%** compared to other methods, with the generated ideas exceeding the acceptance level of top conferences. Human evaluation highlights the practical value of the generated ideas in supporting scientific research while ablation studies further confirm the effectiveness of each component of the workflow.

## 1 INTRODUCTION

The emerging agentic power of Large Language Models (LLMs) Li et al. (2025); Wu et al. (2025); Zhao et al. (2025b) has inspired wide rang of researchers to design LLM agents to automate scientific discovery Wang et al. (2023); Lu et al. (2024); Peng et al. (2025), which is often known as AI scientist systems Yamada et al. (2025); Gottweis et al. (2025); Yu et al. (2024); Qi et al. (2024). Ideation, the ability to generate novel yet feasible research ideas, is arguably one of the most important capabilities, as it shapes the direction of scientific inquiry and influences the course of human progress Coccia (2019); Langley (1987). The ability to generate innovative research ideas has thus become a central focus, with Large Language Models (LLMs) emerging as powerful tools to enhance this process. Recent breakthroughs in LLMs in complex reasoning DeepSeek-AI et al. (2025); Muennighoff et al. (2025); Chen et al. (2025) and world knowledge Yang et al. (2025); Team et al. (2025) have significantly accelerated efforts to harness AI for advancing scientific ideation Si et al. (2024); Su et al. (2024); Pu et al. (2025).

Human scientists have long constructed meaningful relationships between scientific concepts through literature, forming a rich scientific concept network. Previous methods have attempted to capture these relationships using semantic embeddings, focusing on concept similarity. However, these approaches only learn a static representation of each concept, overlooking the nuanced, co-occurrence-based relationships that human scientists build through research. More recent work leveraging LLMs has demonstrated the potential for iterative optimization of scientific ideas by harnessing the vast world knowledge of these models Zheng et al. (2025b); Schmidgall et al.. Yet, these methods fail to tap into the dynamic, evolving nature of the scientific concept network, missing the opportunity to continuously retrieve and integrate the complex, context-dependent relationships between concepts.

Although enabling LLMs to continuously interact with the scientific network is promising, it is nontrivial to achieve this. On one hand, extracting meaningful concept relationships from scientific literature is complex due to the nuanced, context-dependent nature of these connections. On the other hand, enabling LLMs to dynamically interact with this network and incorporate new knowledge throughout the ideation process poses significant difficulties

To construct the scientific concept network, we crawled approximately 100,000 PDFs from the past decade's top 10 AI conferences and analyzed various sections of these papers. Using LLMs, we extracted keywords from the papers and captured the relationships between these keywords as constructed by scientists within each paper. To integrate this scientific keyword network with LLM Agents for generating scientifically grounded ideas and allow LLM agents to optimize their ideas through dynamic interaction with the knowledge base, we propose the **Deep Ideation Framework**. The construction of the scientific network and the Deep Ideation process are shown in Figure 1. First, we construct a scientific network based on keyword co-occurrence within scientific literature and design three key components within the framework: relation analysis, keyword selection, and idea formulation. Secondly, we design a dynamic workflow where the LLM iteratively explores the relationships between existing keywords constructed by previous study within the scientific network, expanding the current set of scientific keywords used to inform the generation of ideas. As the keyword set evolves, the idea proposal is continuously refined and improved. Throughout this iterative process, a idea stack is maintained, providing the LLM with a global perspective of the whole research progress. Finally, we fine-tune the LLM using publicly available review data, enabling it to provide feedback on the generated ideas through a scientifically grounded perspective.

We conducted extensive experiments across four AI research domains to evaluate the performance of our proposed method. The results demonstrate a significant improvement in idea generation, with our approach outperforming the best baseline by an average of 10.25% across four AI domains, reaching the acceptance level of eight out of ten AI conferences. In addition, a human evaluation was conducted to assess the practical impact of our approach, where the generated idea proposals were found to provide genuine inspiration and value to researchers. Furthermore, we conducted ablation studies and a case study, which validated the effectiveness of each module in the Deep Ideation workflow and demonstrated the novelty of the generated ideas.

The key contributions of this work are as follows:

- We collect approximately 100,000 papers from ten major AI conferences over the past decade, constructing a vast scientific concept network based on the co-occurrence and relationships of keywords. This dataset will be made publicly available for the research community to foster further collaboration and exploration.

- We propose the deep-ideation framework, which leverages the scientific concept network to iteratively retrieve and incorporate scientific concept relationships, enabling the generation of high-quality research ideas through an evolutionary search and refinement process.

- We create a review dataset derived from real-world reviewer feedback, and use this dataset to train a critic model which guides the ideation process under expert-level evaluation.

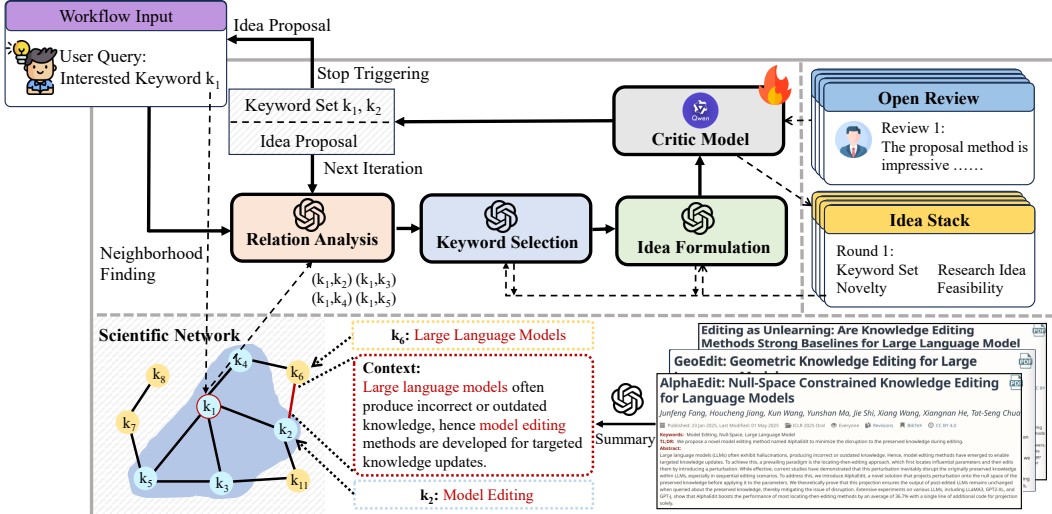

Figure 1: Illustration of the construction of the scientific network and the Deep Ideation process

## 2 PROBLEM FORMULATION

### 2.1 DEFINITION OF SCIENTIFIC NETWORK

Scientific literature forms a vast corpus of interconnected concepts, methodologies, and findings. Organizing this knowledge as a graph Badalyan et al. (2024) efficiently represents complex relationships between scientific ideas Wang et al. (2023); Sourati & Evans (2023), while facilitating easy retrieval and navigation. Keywords, which encapsulate the core themes of a paper, are essential carriers of scientific knowledge. The relationships between co-occurring keywords in individual papers help construct a scientific network, reflecting the interconnections between concepts across the literature.

Formally, let $G = (V, E)$ denote the scientific network, where $V = \{v_1, v_2, \ldots, v_n\}$ represents the set of nodes corresponding to keywords, and $E \subseteq V \times V$ represents the edges between them. An edge $(v_i, v_j) \in E$ exists if keywords $v_i$ and $v_j$ co-occur in at least one paper. The feature $F_{ij}$ of the edge $(v_i, v_j)$ is defined as a function of the relationship between $v_i$ and $v_j$ across all papers in which they both appear. Specifically, let $P_{i,j}$ denote the set of papers in which both keywords $v_i$ and $v_j$ co-occur. Then, the feature $F_{ij}$ of the edge $(v_i, v_j)$ can be represented as:

$$F_{ij} = g\left(\{\text{relation}(v_i, v_j, p) \mid p \in P_{i,j}\}\right),$$

where $\text{relation}(v_i, v_j, p)$ captures the relationship between $v_i$ and $v_j$ in paper $p$, and $g$ is a function that aggregates or processes these relationships to form a meaningful feature representing the connection between the keywords across multiple papers.

### 2.2 FORMULATING RESEARCH IDEATION PROBLEM

Research ideation is a crucial aspect of scientific research Wang et al. (2023); Reddy & Shojaee (2025); Zheng et al. (2025a), where researchers generate and refine novel ideas based on existing knowledge. The effectiveness of scientific ideation depends on how well it leverages prior knowledge to address gaps or challenges in the current research landscape. Therefore, in this paper, we define scientific ideation as the process through which an initial set of keywords, representing key concepts in the research domain, is transformed into an idea proposal that meaningfully synthesizes these concepts in an innovative way.

Formally, let $I = f(K, \Theta, \Psi)$ represent the idea proposal generated by the system, where $K = \{k_1, k_2, \ldots, k_n\}$ is the set of input keywords, $\Theta$ represents the model parameters or system workflow of the ideation system, and $\Psi$ denotes the external knowledge base that informs the ideation process. The idea proposal $I$ is thus a function of the keywords, the system's parameters, and the accessible

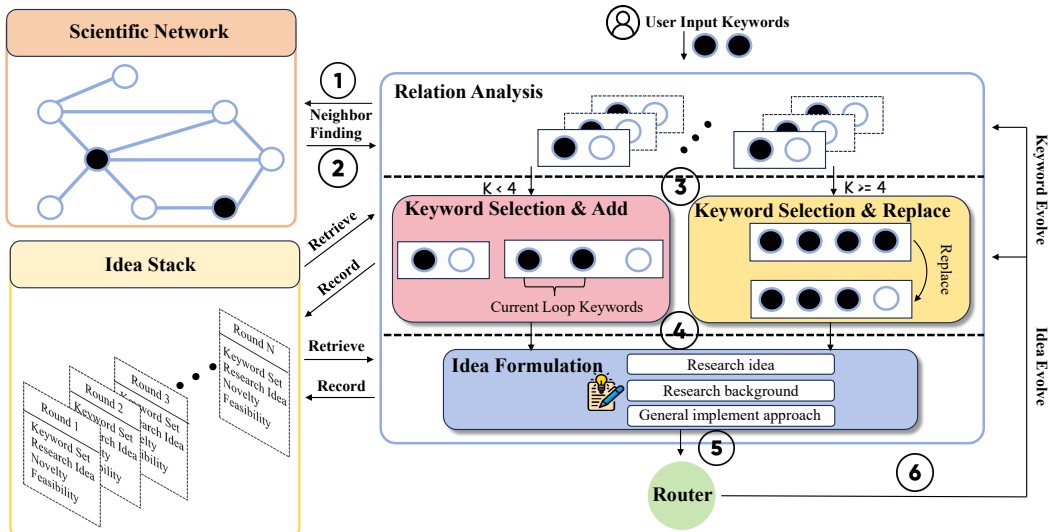

Figure 2: Overview of our Deep Ideation framework. In this figure, we set the maximum size of the keyword set to 4.

external knowledge base, i.e., $I = f(K, \Theta, \Psi)$. To capture the iterative nature of the ideation process, let:

$$I_{t+1} = f(K_t, \Theta_t, \Psi_t), \quad K_{t+1} = \phi(K_t, I_t, \Psi_t),$$

where $I_t$ denotes the idea proposal at iteration $t$, $K_t$ represents the set of keywords at iteration $t$, and $\Psi_t$ represents the external knowledge base at iteration $t$. The function $\phi$ indicates the refinement of the keyword set based on the output of the previous idea proposal and external knowledge. This iterative cycle continues until the generated idea reaches a satisfactory level of novelty and feasibility, or until a stopping criterion is met.

## 3 METHODS

### 3.1 OVERVIEW

To generate scientifically grounded ideas, we propose the Deep Ideation Framework, which integrates a scientific network with LLM Agents for dynamic interaction. The framework operates iteratively, where the LLM queries a constructed scientific network based on keyword co-occurrence within scientific literature. In parallel, the Idea Stack tracks the progression of ideas, offering real-time feedback and guidance on the evolving research proposal, much like how human researchers refine their hypotheses over time through accumulated insights. During each iteration, the framework incorporates an evolving keyword management process Romera-Paredes et al. (2024); Ma et al. (2024), where the LLM is able to iteratively replace keywords within the scientific network. To further refine this process, we introduce a Review Model, trained on publicly available review data, which critically evaluates the novelty and feasibility of the generated ideas, guiding the LLM's ideation process with evaluative feedback. The overall process is illustrated in Figure 2.

### 3.2 KEY COMPONENTS OF DEEP IDEATION FRAMEWORK

This section presents the construction of the scientific network and introduces four key components of the Deep Ideation Framework: the Scientific Network, the Relation Analysis Module, the Keyword Selection Module, and the Idea Formulation Module. We provide the detailed prompts used for the framework in Appendix A.1.

**The Scientific Network** is constructed based on the definition provided in Section 2.1. Initially, the article's title, abstract, and introduction are input into the LLM. Using a "select first, then supplement" approach, the LLM extracts relevant keywords from these sections. These keywords are

treated as nodes in the network. Subsequently, co-occurring keywords within the same paper are connected by edges. The content of each paper defines the relationship between the connected keywords, serving as the feature that characterizes these edges.

**Relation Analysis Module** is responsible for summarizing how the co-occurring papers construct connections between the keywords. Specifically, it analyzes the relationships between keywords and their neighboring terms as established in the literature, capturing the way these terms are linked in the context of scientific research.

**Keyword Selection Module** plays a crucial role in steering the ideation process by selecting the most significant and impactful keywords to expand the initial set. Beyond merely refining the keyword collection, this module actively shapes the direction of the evolving idea, ensuring that it remains focused on the most promising avenues for both novelty and feasibility.

**Idea Formulation Module** addresses a key gap in many existing approaches, which often focus solely on keyword combinations without providing a complete, structured idea proposal Sourati & Evans (2023). This module plays a critical role in synthesizing the selected keywords into a coherent and scientifically grounded idea proposal, transforming a set of keywords into a fully formed concept.

## 3.3 DEEP IDEATION FRAMEWORK

### 3.3.1 EXPLORE

The process begins with an initial set of keywords $K_0 = \{k_1, k_2, \ldots, k_n\}$, which are refined by identifying and analyzing their neighboring terms within the scientific network. To obtain the neighboring keywords, we define $N(K_0)$ as the set of neighboring keywords for all $k_i \in K_0$. Since the number of neighbors for each keyword may be large, we limit the selection to the $m$ neighboring terms, where $m$ is a predefined maximum number. This gives us a set of neighboring keywords for each $k_i$:

$$N(K_0) = \{N(k_1), N(k_2), \ldots, N(k_n)\}$$

Each $N(k_i)$ is limited to the $m$ neighbors. The **Relation Analysis Module** then analyzes the relationships between each pair of selected keywords $(k_i, k_j)$ and their common co-occurrence across multiple papers. Given that multiple papers can share co-occurring keywords, the relationship $R(k_i, k_j)$ between two keywords is derived by considering all the papers where both $k_i$ and $k_j$ appear, represented by $\mathcal{P}(k_i, k_j)$:

$$R(k_i, k_j) = g(k_i, k_j, \mathcal{P}(k_i, k_j))$$

where $\mathcal{P}(k_i, k_j) = \{p_1, p_2, \ldots, p_t\}$ represents the set of papers $p$ that both $k_i$ and $k_j$ co-occur in, and $g$ is a function that aggregate the relationship together.

### 3.3.2 EXPAND

Following the exploration and relation analysis, the **Keyword Selection Module** is tasked with selecting the most significant keyword $k_{\text{new}}$ to add to the current set $K_t$, where $k_{\text{new}} \in N(K_0)$. The selection is based on a comprehensive analysis of the relationship between the new keyword and the existing set of keywords. This new keyword is chosen by evaluating the relationships $R(k_{\text{new}}, k_i)$ for each $k_i \in K_t$, where the relationship between the newly selected keyword and an existing keyword is considered:

$$R(k_{\text{new}}, k_i) = g(k_{\text{new}}, k_i, \mathcal{P}(k_{\text{new}}, k_i))$$

The Keyword Selection Module outputs the selected keyword $k_{\text{new}}$, the reason for the selection (based on its relationship to the current keyword set), and its connection to the existing keyword. The selected keyword is then added to the current keyword set:

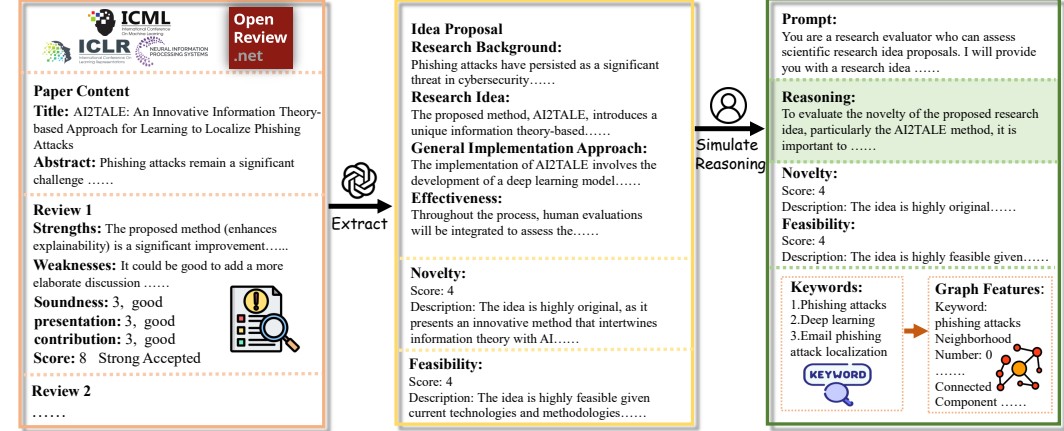

Figure 3: The construction process of the training data for the Review Model

$$K_{t+1} = K_t \cup \{k_{\text{new}}\}$$

Subsequently, this updated set of keywords $K_{t+1}$ and the **Idea Stack**, which contains all previous research iterations (including keyword sets and idea proposals), are input into the **Idea Formulation Module**. The Idea Formulation Module synthesizes the selected keywords into a coherent idea proposal, which includes the research background, research idea, and a general implementation approach. The idea proposal at time $t$ is generated as:

$$P_t = LLM(K_{t+1}, \text{prompt})$$

where the prompt represent the prompt template for idea formulation module. The Idea Stack records each round's progress, tracking keyword evolution, idea development, and evaluations, thus mirroring the iterative nature of human research.

### 3.3.3 EVOLVE

The Evolve Mechanism triggers when the keyword set reaches a predefined length $L_{\max}$. At this point, the focus shifts to evolving the keyword set or the idea proposal. The **Router** determines whether the focus should be on refining the keyword set or on adjusting the idea proposal. The Router decision is formalized as:

$$\text{Next Action} = \begin{cases} \text{Keywords Evolve} & \text{if Router} == \text{Evolve}(K_t) \\ \text{Idea Proposal Evolve} & \text{if Router} == \text{Evolve}(P_t) \end{cases}$$

During the evolution phase, the keywords in $K_t$ are dynamically replaced based on insights from previous iterations. This evolution is represented by:

$$K_{t+1} = (K_t \setminus \{k_{\text{old}}\}) \cup \{k_{\text{new}}\} \quad \text{or} \quad P_{t+1} = LLM(K_{t+1}, \text{prompt})$$

The idea proposal is refined by incorporating new findings and emerging research trends, while the keyword set is updated iteratively to adapt to the evolving research context. This ensures that the generated ideas continue to evolve, progressively becoming more novel and feasible.

### 3.4 CRITIC MODEL

The Critic Model serves as the driving force behind the iterative refinement and evolution of ideas within the Deep Ideation framework. By providing evaluative feedback on the quality of generated idea proposals, it plays a central role in guiding the ideation process through continuous refinement.

This feedback loop ensures that each iteration of the framework is grounded in expert-level evaluation, enabling the system to evolve and generate progressively more innovative and scientifically valid ideas.

While direct use of an LLM for the review process is possible, it falls short in replicating the nuanced evaluative reasoning employed by expert reviewers, generating surface-level assessments but lack the deep, domain-specific understanding. To overcome this, we designed a "Scientific Reasoning Simulation" prompt to enable the LLM to provide review feedback aligned with human reviewers' scientific thinking Zhao et al. (2025a). In this prompt, the LLM is directed to simulate a reviewer's cognitive process, evaluating an idea's novelty and feasibility based on existing research. This simulated reasoning is structured into training data to fine-tune the LLM, aligning its evaluations with peer review standards. Consequently, the review engine provides contextually relevant, scientifically rigorous feedback consistent with expert practices. The process of constructing the training data is referred to in Figure 3

## 4 EXPERIMENTS

### 4.1 EXPERIMENTAL SETUP

**Dataset.** For the dataset, we curated a collection of about 100,000 research papers from major AI conferences over the past decade. These papers were grouped into four categories: DL, NLP, CV, and General AI. The dataset details are provided in the Appendix A.2.

**Baselines.** We compare our approach with several prominent methods in AI-driven scientific discovery, including Sci. Net. Emb. Sourati & Evans (2023), SciMON Wang et al. (2024a), SciAgents Ghafarollahi & Buehler (2025), MOOSE-Chem Yang et al. (2024), Zero-Shot Hypothesis Proposers Qi et al. (2023), ResearchAgent Baek et al. (2024) and papers accepted in the latest year from major AI conferences. More details are presented in Appendix A.3.

**Implementation Details.** In the Deep Ideation framework, we use GPT-4o-mini for all the components, while Qwen3-8B is used for Critic model. We evaluate the novelty and feasibility of the generated ideas using five advanced models: GPT-4o, Gemini-2.5-Flash, Grok-3, DeepSeek-V3.1, and Qwen3-235B-A22B. The final performance score is averaged across these models. More details on the model fine-tuning and evaluation process are provided in Appendix A.4.

### 4.2 EXPERIMENTAL RESULTS

#### 4.2.1 LLM AS JUDGE EVALUATION RESULTS.

| Method | DL | | | NLP | | | CV | | | General AI | | | Overall | | |
|---|---|---|---|---|---|---|---|---|---|---|---|---|---|---|---|
| | Nov. | Fea. | Avg. | Nov. | Fea. | Avg. | Nov. | Fea. | Avg. | Nov. | Fea. | Avg. | Nov. | Fea. | Avg. |
| Accepted Papers | 3.72 | 3.93 | 3.83 | 3.70 | 3.95 | 3.83 | 3.73 | 3.86 | 3.78 | 3.68 | 3.90 | 3.79 | 3.71 | 3.91 | 3.81 |
| Sci. Net. Emb. | 3.34 | 3.53 | 3.44 | 3.24 | 3.61 | 3.43 | 2.64 | 3.44 | 3.04 | 3.26 | 3.57 | 3.42 | 3.12 | 3.53 | 3.33 |
| Scimon | 3.19 | 3.48 | 3.34 | 3.31 | 3.65 | 3.24 | 2.50 | 3.02 | 3.76 | 3.44 | 3.52 | 3.48 | 3.11 | 3.42 | 3.27 |
| SciAgents | 2.93 | 3.63 | 3.28 | 2.86 | 3.69 | 3.28 | 2.73 | 3.65 | 3.19 | 2.75 | 3.27 | 3.01 | 2.82 | 3.46 | 3.14 |
| MOOSE-Chem | 3.53 | 3.33 | 3.43 | 3.43 | 3.22 | 3.33 | 3.34 | 3.21 | 3.28 | 3.46 | 3.07 | 3.27 | 3.44 | 3.21 | 3.33 |
| Zero-Shot HP | 2.80 | 3.65 | 3.23 | 2.73 | 3.46 | 3.10 | 2.78 | 3.61 | 3.20 | 2.81 | 3.51 | 3.16 | 2.78 | 3.57 | 3.18 |
| ResearchAgent | 3.38 | 3.22 | 3.30 | 3.54 | 3.33 | 3.44 | 3.48 | 3.28 | 3.38 | 3.43 | 3.25 | 3.34 | 3.46 | 3.47 | 3.47 |
| Deep Ideation | 3.79 | 3.86 | 3.83 | 3.70 | 3.92 | 3.81 | 3.81 | 3.89 | 3.85 | 3.73 | 3.90 | 3.82 | 3.76 | 3.88 | 3.82 |
| Improvement↑ | 7.37% | 5.75% | 10.92% | 4.52% | 6.23% | 9.48% | 9.48% | 6.58% | 13.91% | 7.80% | 9.24% | 9.48% | 8.67% | 8.68% | 10.25% |

Table 1: Performance of Deep Ideation with LLM as Judge compared to Baselines across Different AI Domains. Bold and underline indicate the best and second best performance(except Accepted Papers).

As shown in Table 1, Deep Ideation demonstrates a significant improvement across all domains. Specifically, Deep Ideation shows a substantial increase in the Avg. score: 10.92% in the DL domain, 9.48% in the NLP domain, 13.91% in the CV domain, and 9.48% in the General AI domain compared to the best-performing baseline. This performance is a direct result of our approach's ability to generate scientifically novel ideas that are both technically feasible and relevant to the research landscape. Additionally, Deep Ideation surpasses the level of many AI conference accepted papers in several domains, further highlighting the robustness and effectiveness of our approach.

The impact and analysis of different parameter settings on the final performance are presented in Appendix A.5.

### 4.2.2 HUMAN EVALUATION RESULT

To further evaluate the effectiveness of the Deep Ideation framework, a human evaluation is conducted involving 54 researchers, and details are provided in the Appendix A.6). The human evaluation results are presented in Table 2.

| Method | DL | | | NLP | | | CV | | | General AI | | | Overall | | |
|---|---|---|---|---|---|---|---|---|---|---|---|---|---|---|---|
| | Nov. | Fea. | Avg. | Nov. | Fea. | Avg. | Nov. | Fea. | Avg. | Nov. | Fea. | Avg. | Nov. | Fea. | Avg. |
| Accepted Papers | 3.57 | 3.74 | 3.66 | 3.63 | 3.80 | 3.72 | 3.70 | 3.56 | 3.63 | 3.54 | 3.72 | 3.63 | 3.61 | 3.71 | 3.66 |
| Sci. Net. Emb. | 3.15 | 3.24 | 3.19 | 3.11 | 3.24 | 3.18 | 2.94 | 3.11 | 3.03 | 3.22 | 3.35 | 3.29 | 3.11 | 3.24 | 3.18 |
| Scimon | 2.94 | 2.85 | 2.90 | 2.96 | 2.91 | 2.94 | 3.15 | 3.17 | 3.16 | 2.89 | 3.24 | 3.07 | 2.99 | 3.04 | 3.02 |
| SciAgents | 2.87 | 3.20 | 3.04 | 3.11 | 3.15 | 3.13 | 3.24 | 3.26 | 3.25 | 2.72 | 3.09 | 2.91 | 2.99 | 3.18 | 3.09 |
| MOOSE-Chem | 3.43 | 3.20 | 3.32 | 3.07 | 3.19 | 3.13 | 3.35 | 3.38 | 3.37 | 3.35 | 3.48 | 3.39 | 3.30 | 3.31 | 3.31 |
| Zero-Shot HP | 2.72 | 2.93 | 2.83 | 2.64 | 3.22 | 2.93 | 3.22 | 3.31 | 3.27 | 2.76 | 3.11 | 2.94 | 2.84 | 3.14 | 2.99 |
| ResearchAgent | 3.22 | 3.26 | 3.24 | 3.24 | 3.41 | 3.33 | 3.11 | 3.19 | 3.15 | 3.43 | 3.63 | 3.53 | 3.25 | 3.37 | 3.31 |
| Deep Ideation | 3.65 | 3.72 | 3.69 | 3.61 | 3.76 | 3.69 | 3.74 | 3.57 | 3.66 | 3.61 | 3.81 | 3.71 | 3.65 | 3.72 | 3.69 |

Table 2: Performance of Deep Ideation with human evaluation compared to Baselines across Different AI Domains. Bold and underline indicate the best and second best performance(except Accepted Papers).

The results shows that our method consistently outperformed the best baseline, demonstrating Deep Ideation's ability to generate more valuable, innovative, and well-structured ideas. One participant notes that the ideas generated by Deep Ideation are "highly innovative and grounded in existing scientific knowledge, providing fresh perspectives on complex problems."

### 4.3 ABLATION STUDY OF DEEP IDEATION

To validate the effectiveness of each module in Deep Ideation, we conducted an ablation study. The results of this experiment are presented in Table 3.

| Method | DL | | | NLP | | | CV | | | General AI | | | Overall | | |
|---|---|---|---|---|---|---|---|---|---|---|---|---|---|---|---|
| | Nov. | Fea. | Avg. | Nov. | Fea. | Avg. | Nov. | Fea. | Avg. | Nov. | Fea. | Avg. | Nov. | Fea. | Avg. |
| Deep Ideation(full) | 3.79 | 3.86 | 3.83 | 3.70 | 3.92 | 3.81 | 3.81 | 3.89 | 3.85 | 3.73 | 3.90 | 3.82 | 3.76 | 3.88 | 3.82 |
| w/o Evolve | 3.74 | 3.68 | 3.71 | 3.59 | 3.82 | 3.71 | 3.64 | 3.80 | 3.72 | 3.66 | 3.74 | 3.70 | 3.66 | 3.76 | 3.71 |
| w/o Critic Model. | 3.61 | 3.64 | 3.63 | 3.43 | 3.63 | 3.53 | 3.58 | 3.63 | 3.61 | 3.52 | 3.62 | 3.57 | 3.54 | 3.63 | 3.59 |

Table 3: Ablation study of Deep Ideation across different AI domains.

**Effectiveness of Evolve Mechanism.** As shown in Table 3, removing the evolve mechanism (w/o Evolve) results in a noticeable decline in performance across all domains. This degradation stems from the inability of the agent to adaptively replace unsuitable keywords based on review feedback. Without this iterative evolution, the generated ideas become static, failing to dynamically adjust to the shifting scientific context and explore deeper insights, leading to lower-quality proposals.

**Effectiveness of Critic Model.** Without the critic model, the idea generation process lacks evaluative guidance, causing the evolution of ideas to become directionless and blind. This absence of structured feedback leads the agent to explore ideas in an uncoordinated manner, without any clear alignment to scientific objectives or constraints. As a result, the generated ideas become disconnected from the underlying scientific context, which significantly undermines their novelty and feasibility.

### 4.4 CASE STUDY

These case studies in Figure 4 demonstrate Deep Ideation's ability to generate novel solutions. In multi-face forgery detection and deepfake prevention, the dual-task model innovatively combines face detection and segmentation with reinforcement learning, dynamically selecting the best strategies in real-time. In adaptive self-evolution and preference alignment, incorporating Noise Contrastive Estimation (NCE) into reinforcement learning offers a novel approach to overcoming biases

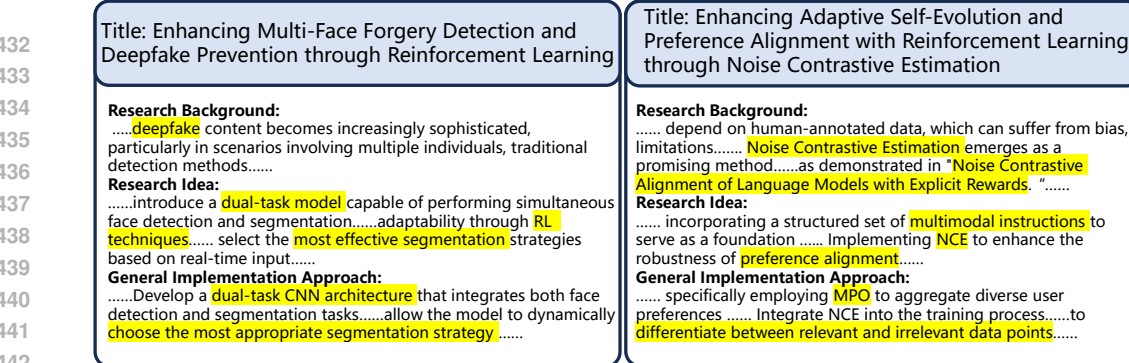

Figure 4: A case study of idea proposal generated by Deep Ideation.

in human-annotated data, improving preference robustness. Additionally, in the right case, the idea proposal cites a relevant paper, demonstrating how Deep Ideation effectively incorporates existing research to refine and enhance its generated ideas.

## 5 RELATED WORKS

### 5.1 LLMs FOR SCIENTIFIC RESEARCH

In recent years, the application of LLMs to scientific research has garnered significant attention Yamada et al. (2025); Swanson et al. (2025); Shi et al. (2023); Hsu et al. (2024) and a number of task-specific systems have emerged. ResearchAgent Baek et al. (2024) combines LLMs with symbolic knowledge graphs to iteratively propose, refine, and simulate experimental designs, while AutoSurvey Wang et al. (2024b) automates the generation of literature surveys by chaining retrieval, clustering, and multi-agent writing modules. AlphaEvolve Novikov et al. (2025) takes a step further by using an LLM-guided genetic algorithm to evolve code-level hypotheses and make new discoveries in computational research.

### 5.2 LLMs FOR SCIENTIFIC IDEATION

Recent advancements in LLM-based scientific ideation have focused on utilizing the powerful capabilities of LLMs to autonomously iterate and generate new scientific research ideas. Approaches like SciMON Wang et al. (2024a) and ResearchAgent Baek et al. (2024) have utilized iterative processes, where LLMs refine hypotheses by continuously incorporating new literature, improving novelty and relevance through real-time retrieval and semantic novelty maximization. Multi-agent systems, such as SciAgents Ghafarollahi & Buehler (2025), further advanced this by automating hypothesis validation and refinement, showing improvements in both novelty and feasibility. Additionally, MOOSE-Chem Yang et al. (2024) demonstrated LLMs' potential to rediscover hidden knowledge by mining knowledge graphs of patents, while Qi et al. Qi et al. (2023) showcased the zero-shot capability of LLMs in generating valid hypotheses without the need for explicit examples. CycleResearcher Weng et al. (2024) introduced a self-supervised feedback loop, incorporating automated reviews to iteratively enhance the quality of generated hypotheses.

## 6 CONCLUSION

We presented the Deep-Ideation framework, which integrates LLMs with scientific networks to generate novel and scientifically grounded research ideas. By leveraging the relationships between keywords in scientific literature, our method ensures that generated ideas are both innovative and anchored in existing knowledge. The iterative workflow, enhanced by the Idea Stack, enables continuous idea refinement, mirroring the cognitive process of human researchers. Additionally, the review model, trained on real-world feedback, provides critical evaluative input to ensure that the ideas are novel and feasible. Our experiments across various AI research domains demonstrate significant improvements in both novelty and feasibility, highlighting the effectiveness of our approach.

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

# A APPENDIX

## A.1 PROMPT FOR DEEP IDEATION

### A.1.1 PROMPT OF RELATION ANALYSIS

**Prompt:**

You are a research assistant tasked with analyzing academic papers.

I will provide you with two keywords and a paper in which both keywords appear. Your task is to carefully read the paper and explain how the paper constructs the connection between these two keywords.

Here are the inputs:

- Keyword 1: {keyword1}

- Keyword 2: {keyword2}

Paper:

Title: title

Abstract: abstract

Introduction: introduction

Please explain how this paper constructs the connection between the two keywords, limited to 2-4 sentences.

### A.1.2 PROMPT OF KEYWORD SELECTION

**Prompt:**

You are a research assistant who can help expand the current set of research keywords.

I will provide you with:

1. A "Idea Stack" that records the entire research progress across all previous rounds. Each round in the Idea Stack contains:

- The current set of keywords for each round.

- The current research idea for each round.

- Novelty and feasibility scores and their descriptions of current research idea proposal for each round.

- The addition(or replacement) of new keywords in this round, including which keyword was added(or replaced) and the reason for its addition(or replacement).

- The refined idea after the addition of the new keyword, guided by the research progress recorded in the Idea Stack.

The Full Idea Stack represents an iterative process of refining the research idea. Each round in the Idea Stack represents a research part of the research journey, reflecting the evolution of the research direction, where new keywords and concepts are integrated based on the previous evaluations. Use the full Idea Stack to:

- Understand the evolution of the research direction across rounds, and how past additions and adjustments of keywords set has influenced the current idea.

- Avoid selecting keywords that are redundant or overly similar to past directions.

- Ensure that the selected new keyword logically builds upon the research progress recorded in the Idea Stack and contributes to the overall improvement of the research idea.

2. A list of new candidate keywords. Each of them has a relationship with one keyword from the current keywords set; through this relationship, the new keyword becomes connected to the whole current keyword set.

- Additionally, each candidate keyword may has an associated shortest path length to the other existing keywords (different of the keyword they have a relationship with) in the current set if they are connected to each other. The shortest path indicates how distant the new keyword is from the current keyword set in the scientific network. The shorter the path, the closer the keyword is related to the existing keywords.

Your task:

- Carefully analyze the entire Idea Stack, taking into account novelty, feasibility, and past keyword selections and reasons.

- If the novelty of the current research idea is insufficient, prefer new keywords that could improve novelty of the latest research idea. You may consider new keywords with a longer shortest path as a potential way to introduce more diverse concepts and increase novelty. However, novelty improvement should not rely solely on the shortest path—ensure the new keyword also aligns with the relevance and focus of the current research.

- If the feasibility is weak, prefer new keywords that could improve feasibility of the latest research idea. You may consider new keywords with a shorter shortest path to ensure the idea remains grounded in existing knowledge and is practical to implement. Again, feasibility improvement should not rely solely on the shortest path—balance it with other considerations to ensure the keyword strengthens the idea's feasibility.

- Avoid choosing keywords that would make the research direction redundant with previous selections.

- Select ONE NEW keyword that has the highest potential to enhance the research idea in light of the Idea Stack.

- When you output the result, specify which existing keyword the new keyword is connected to.

- Additionally, briefly explain the reason for choosing this keyword, emphasizing how it will help improve the current research idea and how it builds on the historical progress in the Idea Stack.

The Idea Stack(contains full history of research progress): {idea_stack}

The following are new candidate keywords and their relationships to specific keywords in the current set, including the shortest path between the candidate keywords and existing keywords:

{candidate_keywords_and_relationships}

Please output your choice in the following format:

NEW_KEYWORD: (the new keyword you selected)

CONNECTED_TO: (the specific keyword in the current set that relates to the new keyword)

REASON_FOR_SELECTION: (a brief explanation for why this keyword was selected, considering the improvement it will bring to current research idea and how it builds on the historical progress in the Idea Stack)

### A.1.3 PROMPT OF KEYWORD REPLACEMENT

**Prompt:**

You are a research assistant who helps refine the current set of research keywords by replacing certain existing keywords.

I will provide you with:

1. The whole keywords set in the current round.

2. The flexible keywords set, which is a subset of the whole keywords set. These flexible keywords can be replaced with new candidate keywords.

3. A "Idea Stack" that records the entire research progress across all previous rounds. Each round in the Idea Stack contains:

- The current set of keywords for each round.

- The current research idea for each round.

- Novelty and feasibility scores and their descriptions of the current research idea proposal for each round.

- The addition(or replacement) of new keywords in this round, including which keyword was added(or replaced) and the reason for its addition(or replacement).

- The refined idea after replacing the keyword, guided by the research progress recorded in the Idea Stack.

The Full Idea Stack represents an iterative process of refining the research idea. Each round in the Idea Stack reflects a research part of the journey, showing how the research direction evolves with the integration of new and refined keywords. Use the full Idea Stack to:

- Understand the evolution of the research direction across rounds, and how previous keyword replacements have influenced the current idea.

- Avoid replacing keywords that make the research direction redundant or overly similar to past directions.

- Ensure that the selected replacement keyword logically builds upon the research progress recorded in the Idea Stack and improves the overall research idea.

4. A list of candidate replacement keywords. Each of them has a relationship with one keyword from the current keywords set; through this relationship, the new replacement keyword becomes connected to the whole current keyword set. Each candidate replacement keyword can only replace one of the flexible keywords set.

- Additionally, each candidate replacement keyword has an associated shortest path length to the other existing keywords in the whole keywords set (different of the keyword they have a relationship with) if they are connected to each other. The shortest path indicates how closely the replacement keyword is related to the existing keyword set. The shorter the path, the more relevant the keyword is to the existing ideas.

Your task:

- Carefully analyze the entire Idea Stack, taking into account the novelty, feasibility, and past keyword replacements and reasons.

- If the novelty of the current research idea is insufficient, prefer replacing a flexible keyword with one that could improve novelty. You may consider keywords with a longer shortest path to introduce more diverse concepts and improve novelty, but novelty improvement should not rely solely on path length—ensure that the replacement keyword aligns with the research focus.

- If the feasibility is weak, replace a flexible keyword with one that could enhance feasibility. Keywords with a shorter shortest path to existing keywords could be preferred to ensure the idea is grounded in practical, existing knowledge.

- Avoid replacing a keyword with one that makes the research direction redundant or conflicts with past research directions.

- Select ONE flexible keyword to replace with a new replacement keyword that holds the highest potential to improve the research idea based on the Idea Stack.

- When you output the result, specify which existing keyword the replacement keyword is connected to, which flexible keyword was replaced and the reason for its replacement, ensuring that the replacement strengthens the research idea and builds upon the historical progress.

The whole keywords set in the current round is:

{keywords}

The flexible keywords set of which keywords can be replaced is: {flexible_keywords}

The Idea Stack (contains full history of research progress): {idea_stack}

The following are new candidate replacement keywords and their relationships to specific keywords in the current set, including the shortest path between the candidate replacement keywords and existing keywords:

{candidate_keywords_and_relationships}

Please output your choice in the following format:

REPLACEMENT_KEYWORD: (the new keyword you selected)

CONNECTED_TO: (the specific keyword in the current set that relates to the replacement keyword)

REPLACED_KEYWORD: (the flexible existing keyword that you replaced)

REASON_FOR_REPLACEMENT: (a brief explanation of why this keyword was replaced, how it improves the research idea, and how it builds upon the research progress in the Idea Stack)

### A.1.4 PROMPT OF IDEA FORMULATION

**Prompt:**

You are a research assistant tasked with generating a novel scientific research idea proposal.

I will provide you with:

1. A "Idea Stack" that records the entire research progress across all previous rounds. Each round in the Idea Stack contains:

- The current set of keywords for each round.

- The current research idea for each round.

- Novelty and feasibility scores and their descriptions of current research idea proposal for each round.

- The addition of new keywords in this round, including which keyword was added and the reason for its addition.

- The refined idea after the addition of the new keyword, guided by the research progress recorded in the Idea Stack.

The Full Idea Stack represents an iterative process of refining the research idea. Each round in the Idea Stack represents a research part of the research journey, reflecting the evolution of the research direction, where new keywords and concepts are integrated based on the previous evaluations. Use the full Idea Stack to:

- Understand the evolution of the research direction across rounds, and how previous additions and adjustments have influenced the current idea.

- Ensure that the new research idea proposal builds upon this iterative process and integrates the new keywords in a coherent way.

- Avoid selecting ideas that are redundant or overly similar to past directions and make sure to contribute something novel.

2. A set of keywords. Your task is to:

- Combine these keywords into a coherent and innovative scientific research idea, using the guidance of the Idea Stack to ensure that the new idea builds logically upon the research progress recorded in the Idea Stack.

- Develop this idea into a research idea proposal that includes:

- The research background: An overview of the research context and importance.

- The research idea: A clear description of the novel idea.

- A general implementation approach: A brief explanation of how the idea can be practically implemented.

Here are the keywords: {keywords}

The Idea Stack (contains the history of all rounds' research progress): {status_bar}

Please output your idea proposal, which should include the research background, research idea, and a general implementation approach. Ensure that the proposal is aligned with the overall research progress recorded in the Idea Stack and effectively integrates the new keywords.

### A.1.5 PROMPT OF REVIEW MODEL

**Prompt:**

You are a research evaluator who can assess scientific research idea proposals.

I will provide you with an idea proposal, the keywords associated with the proposal, the graph features that emerge when these keywords are considered as nodes in a scientific network

- The scientific network is constructed based on the co-occurrence of keywords in scientific papers. Each keyword is represented as a node, and an edge exists between two nodes if the corresponding keywords have appeared together in the same paper.

- The graph features include the following information .

- Neighbor count: Indicates how many other keywords each keyword is directly connected to.

- Connectivity: Indicates whether the keywords of this idea proposal are connected on the graph.

- Shortest paths: Represents the shortest paths of each pair of keywords in this idea proposal on the scientific network.

Your task is to evaluate the research idea proposal along two dimensions:

1. Novelty (1–5): How original and innovative the idea is compared to existing research.

- 5: Extremely novel and groundbreaking. The idea introduces new, unexplored concepts or radically shifts the direction of the field.

- 4: Highly original. The idea is new and innovative but may still be building upon existing concepts or research.

- 3: Moderately original. The idea brings some new insights but is similar to existing work or follows well-established concepts.

- 2: Slightly original. The idea offers minor variations or incremental improvements to existing research but lacks substantial novelty.

- 1: Not original. The idea closely resembles existing research with little to no innovation.

2. Feasibility (1–5): How realistic and practical the idea is to implement in current scientific and technological conditions.

- 5: Fully feasible. The idea can be realistically executed with existing methods, data, and resources, and the plan for implementation is clear and practical.

- 4: Highly feasible. The idea is feasible with current technologies but may require some advancements or additional resources.

- 3: Moderately feasible. The idea faces significant practical challenges, requiring considerable advancements in technology or data.

- 2: Slightly feasible. The idea is difficult to implement with current resources and would need significant breakthroughs.

- 1: Not feasible. The idea is impractical and unlikely to be implemented with current technologies or methods.

Please consider the keyword network features (neighbor count, connectivity, shortest paths) in your evaluation to understand the research idea's potential impact, relevance, and connectivity within the scientific field.

The research idea proposal is: {research_idea}

The keywords in this idea proposal are: {keywords}

The graph-based features of these keywords: {graph_features}

Please output your evaluation strictly in the following format:

Novelty Score and Description: (give a score from 1–5 and a short description of novelty)

Feasibility Score and Description: (give a score from 1–5 and a short description of feasibility)

### A.1.6  PROMPT OF ROUTER

**Prompt:**

You are a decision-making assistant helping to determine the next step in the research workflow. You need to choose whether the next step should be "keyword replacement" or "idea proposal rewriting".

Here is the information provided:

1. Research Idea: The current research idea proposal.

2. Keywords: The current set of research keywords.

3. Novelty Score and Description: The novelty score and its corresponding description of the current research idea.

4. Feasibility Score and Description: The feasibility score and its corresponding description of the current research idea.

Decision Logic: - Perform keyword replacement if:

- The novelty and/or feasibility of the current research idea are insufficient, but the current set of keywords is of high quality. In this case, the problem likely lies with the research idea proposal itself, which can be improved by refining the keywords.

- Rewrite the idea proposal if:

- The keywords set are of high quality, but the idea proposal itself is poorly written. This suggests that the research idea proposal needs improvement to better reflect the potential of the current keywords.

Instructions:

- Review the novelty and feasibility scores to assess the quality of the current research idea.

- If the keywords are strong but the idea proposal is weak, then consider idea proposal rewriting.

- If the novelty or feasibility scores are low, and improving the keywords is necessary, then choose keyword replacement.

Output Format:

Please output your decision as one of the following actions:

1. Keyword_Replacement: If you think the next step should focus on refining the keywords for improving novelty or feasibility.

2. Idea_Rewrite: If you think the next step should focus on reworking the research idea itself to enhance its overall quality.

Current Data:

- Research Idea: {research_idea}

- Keywords: {keywords}

- Novelty Score and Description: {novelty_score_desc}

- Feasibility Score and Description: {feasibility_score_desc}

Please output your choice and briefly explain why you think this is the best next step for improving the research process. Your output should strictly follow this format:

ACTION: (either "Keyword_Replacement" or "Idea_Rewrite") REASON: (a brief explanation of why you chose this action based on the provided data)

### A.1.7 PROMPT OF KEYWORD REPLACE

**Prompt:**

You are a research assistant who helps refine the current set of research keywords by replacing certain existing keywords.

I will provide you with:

1. The whole keywords set in the current round.

2. The flexible keywords set, which is a subset of the whole keywords set. These flexible keywords can be replaced with new candidate keywords.

3. A "Idea Stack" that records the entire research progress across all previous rounds. Each round in the Idea Stack contains:

- The current set of keywords for each round.

- The current research idea for each round.

- Novelty and feasibility scores and their descriptions of the current research idea proposal for each round.

- The addition(or replacement) of new keywords in this round, including which keyword was added(or replaced) and the reason for its addition(or replacement).

- The refined idea after replacing the keyword, guided by the research progress recorded in the Idea Stack.

The Full Idea Stack represents an iterative process of refining the research idea. Each round in the Idea Stack reflects a research part of the journey, showing how the research direction evolves with the integration of new and refined keywords. Use the full Idea Stack to:

- Understand the evolution of the research direction across rounds, and how previous keyword replacements have influenced the current idea.

- Avoid replacing keywords that make the research direction redundant or overly similar to past directions.

- Ensure that the selected replacement keyword logically builds upon the research progress recorded in the Idea Stack and improves the overall research idea.

4. A list of candidate replacement keywords. Each of them has a relationship with one keyword from the current keywords set; through this relationship, the new replacement keyword becomes connected to the whole current keyword set. Each candidate replacement keyword can only replace one of the flexible keywords set.

- Additionally, each candidate replacement keyword has an associated shortest path length to the other existing keywords in the whole keywords set (different of the keyword they have a relationship with) if they are connected to each other. The shortest path indicates how closely the replacement keyword is related to the existing keyword set. The shorter the path, the more relevant the keyword is to the existing ideas.

Your task:

- Carefully analyze the entire Idea Stack, taking into account the novelty, feasibility, and past keyword replacements and reasons.

- If the novelty of the current research idea is insufficient, prefer replacing a flexible keyword with one that could improve novelty. You may consider keywords with a longer shortest path to introduce more diverse concepts and improve novelty, but novelty improvement should not rely solely on path length—ensure that the replacement keyword aligns with the research focus.

- If the feasibility is weak, replace a flexible keyword with one that could enhance feasibility. Keywords with a shorter shortest path to existing keywords could be preferred to ensure the idea is grounded in practical, existing knowledge.

- Avoid replacing a keyword with one that makes the research direction redundant or conflicts with past research directions.

- Select ONE flexible keyword to replace with a new replacement keyword that holds the highest potential to improve the research idea based on the Idea Stack.

- When you output the result, specify which existing keyword the replacement keyword is connected to, which flexible keyword was replaced and the reason for its replacement, ensuring that the replacement strengthens the research idea and builds upon the historical progress.

The whole keywords set in the current round is:

{keywords}

The flexible keywords set of which keywords can be replaced is:

{flexible_keywords}

The Idea Stack (contains full history of research progress):

{status_bar}

The following are new candidate replacement keywords and their relationships to specific keywords in the current set, including the shortest path between the candidate replacement keywords and existing keywords:

{candidate_keywords_and_relationships}

Please output your choice in the following format:

REPLACEMENT_KEYWORD: (the new keyword you selected)

CONNECTED_TO: (the specific keyword in the current set that relates to the replacement keyword)

REPLACED_KEYWORD: (the flexible existing keyword that you replaced)

REASON_FOR_REPLACEMENT: (a brief explanation of why this keyword was replaced, how it improves the research idea, and how it builds upon the research progress in the Idea Stack)

### A.2 DATASET DETAILS

The dataset consists of 107,443 research papers sourced from major AI conferences over the past decade, including ICLR, NeurIPS, ICML, ACL, NAACL, CVPR, ICCV, AAAI, and IJCAI. These papers were grouped into four categories:

- DL (short for Deep Learning): ICLR, NeurIPS, ICML

- NLP (short for Natural Language Process): ACL, NAACL

- CV (Computer Vision): CVPR, ICCV

- General AI: AAAI, IJCAI

From each paper, 3-4 keywords were extracted, forming the basis of a scientific network constructed from keyword co-occurrence. During the idea proposal generation process, an initial keyword was selected from a specific domain. The resulting idea proposal was then classified according to the domain from which the initial keyword was drawn.

### A.3 Baseline Methods

We benchmark our approach against several prominent methods in AI-driven scientific discovery:

- **Sci. Net. Emb.:** Sourati & Evans (2023) This method integrates human expertise into AI models to enhance predictions of future scientific breakthroughs, particularly in data-scarce contexts. By considering the distribution of human expertise, it improves AI-driven predictions beyond traditional research content.

- **SciMON:** Wang et al. (2024a) SciMON focuses on optimizing neural language models for novelty. It iteratively refines generated hypotheses by comparing them with existing literature, aiming to improve both the technical depth and originality of the generated ideas.

- **SciAgents:** Ghafarollahi & Buehler (2025) This method employs ontological knowledge graphs and multi-agent systems to autonomously generate and refine hypotheses. By uncovering interdisciplinary connections, it accelerates material discovery and fosters new research avenues.

- **MOOSE-Chem:** Yang et al. (2024) MOOSE-Chem applies a structured framework to generate hypotheses in chemistry. It demonstrates the potential of LLMs in rediscovering scientifically valuable insights and advancing the hypothesis generation process in the field of chemistry.

- **Zero-Shot Hypothesis Proposers:** Qi et al. (2023) This method explores the ability of LLMs to propose valid hypotheses without prior fine-tuning. It showcases the capability of LLMs to generate novel scientific ideas from unseen literature, pushing the boundaries of zero-shot hypothesis generation.

- **ResearchAgent:** Baek et al. (2024) ResearchAgent combines iterative idea generation with LLM-based review agents to refine scientific proposals. It represents a comprehensive approach to supporting researchers in the ideation process, enhancing both the creativity and rigor of generated ideas.

- **Accepted Papers:** We also include the latest accepted papers from major AI conferences as baselines. Here, we input the title, abstract, and introduction of each paper into the model and asked it to organize the content into an idea proposal format.

### A.4 Evaluation Details and Model Fine-Tuning

In the Deep-Ideation workflow, GPT-4o-mini is used as the backbone model for the modules of Relation Summary, Keyword Selection, Keyword Replace, and Idea Writing. For the Idea Reviewing module, Qwen3-8B is used, which is fine-tuned with Low-Rank Adaptation (LoRA) on a training dataset of 4278 examples. This fine-tuning process enhances the model's ability to evaluate the novelty and feasibility of the generated ideas.

To assess the quality of the generated ideas, five advanced large language models are used for evaluation: GPT-4o, Gemini-2.5-Flash, Grok-3, DeepSeek-V3.1, and Qwen3-235B-A22B. The final performance scores are derived by averaging the results across these models, ensuring a robust and comprehensive evaluation of the ideas' quality.

### A.5 Analysis

In this section, we analyzed key elements of the Deep Ideation framework. Max neighborhood size controls the breadth of knowledge sampled for each keyword, while max keyword set size defines the number of keywords used to generate the idea proposal. The results are shown in Figure A.5.

#### A.5.1 Effect of max neighborhood size.

As shown in Figure A.5 (left), Deep Ideation performs best when the maximum neighborhood size is set to 12. When smaller, the limited scientific knowledge surrounding each keyword restricts the agent's ability to capture comprehensive insights, diminishing the quality and depth of the final idea proposals. Conversely, increasing the neighborhood size beyond 12 expands the agent's knowledge boundary, but may lead to information overload, making it difficult for the agent to prioritize the most valuable insights and causing the focus of the generated ideas to become diluted.

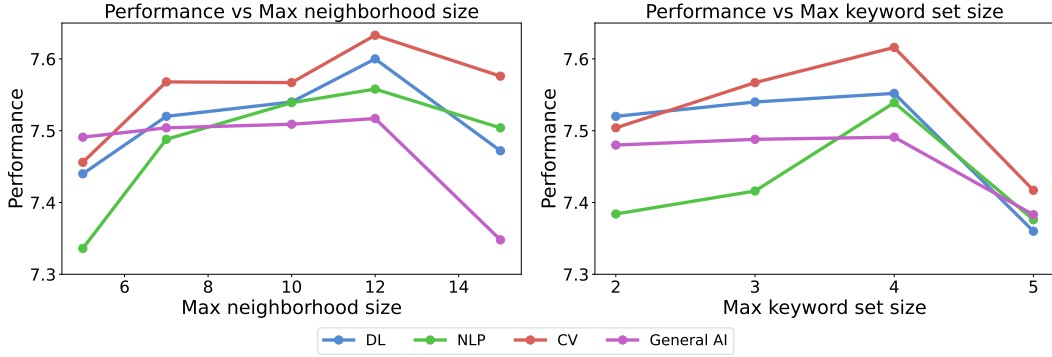

Figure A.5: Effect of max neighborhood size and max keyword set size, where performance is the sum of novelty and feasibility.

### A.5.2 EFFECT OF MAX KEYWORD SET SIZE

Figure A.5 (right) illustrates that, when the keyword set size is small, the knowledge breadth and diversity of the final idea proposal are limited, which results in ideas that lack innovation and depth, often failing to address the complexities of scientific problems. In contrast, increasing the keyword set size too much leads to overly complex relationships between the keywords, causing the ideas to become disjointed or unnatural, with forced connections that undermine clarity. However, when the keyword set size is set to 4, the performance improves significantly, indicating that a balanced set allows the agent to capture sufficient diversity and depth while keeping the generated ideas focused and logically connected.

### A.6 HUMAN EVALUATION DETAILS

Human evaluation is essential for assessing the real-world applicability and impact of generated ideas, ensuring that they meet the expectations of domain experts. In this study, participants were asked to assess each idea proposal across two dimensions: novelty and feasibility. Additionally, they were required to write a brief description for a select number of idea proposals that particularly caught their interest.

### A.7 THE USE OF LARGE LANGUAGE MODELS

In this work, only the grammar correction and sentence-level refinement of the manuscript were carried out using a large language model (LLM).

