# OpenReview forum: "Deep Ideation: Designing LLM Agents to Generate Novel Research Ideas on Scientific Concept Network"
_ICLR.cc/2026/Conference — ICLR 2026 Conference Withdrawn Submission_

### Official Review · Reviewer_hrL1 · 2025-10-29

**Soundness:** 2
**Presentation:** 2
**Contribution:** 2
**Rating:** 2
**Confidence:** 4

**Summary:**

This paper proposes Deep Ideation, a framework that integrates LLMs with a scientific concept network to generate novel and feasible research ideas. The system constructs a concept-relation graph via keyword co-occurrence and contextual relationships, and conducts an explore-expand-evolve workflow. An Idea Stack maintains iterative research progress, while a Critic Model, fine-tuned on real-world review data, provides feedback on novelty and feasibility. Experiments across four domains show about 10 % gains over baselines such as SciMON, ResearchAgent, and MOOSE-Chem by both LLM-judge and human evaluations.

**Strengths:**

(1) The work moves beyond static keyword or embedding-based ideation by dynamically retrieving and composing concept relations from a curated scientific network.

(2) Fine-tuning on real review text gives a realistic feedback signal for novelty and feasibility assessment.

(3) The released concept network and dataset could support further AI-for-Science research.

**Weaknesses:**

(1) The paper does not analyze whether the training corpus, review data, and accepted-paper references pre- or post-date the LLMs’ training cut-off—important for judging fairness and novelty.

(2) Section 4.2.2 lists 54 researchers but omits selection criteria, expertise, or calibration details.

(3) Novelty/feasibility scores are subjective; the paper reports no inter-rater agreement between human evaluators, nor consistency among different LLM judges.

(4) Only Evolve and Critic modules are tested; other components remain unanalyzed.

(5) Occasional redundancy and verbosity, e.g. Appendix A.1.3 and A.1.7 contain essentially identical prompts for keyword replacement, suggesting redundancy or editing oversight.

**Questions:**

- How is the edge feature F_{ij} concretely computed from multiple papers?

- Please clarify whether any papers or reviews used in training or evaluation post-date the LLMs’ own data cut-off.

- How were the 54 human evaluators selected, and how consistent were their scores with LLM-based evaluations?

- Did you measure agreement among different LLM judges (e.g., GPT-4o vs Gemini 2.5 Flash)?

- What is the computational cost and scalability of the system?

- How is the stopping criterion in the Evolve phase determined?

---

### Official Review · Reviewer_Dots · 2025-11-01

**Soundness:** 3
**Presentation:** 2
**Contribution:** 2
**Rating:** 2
**Confidence:** 4

**Summary:**

The paper proposes the Deep Ideation framework, which uses a concept network to iteratively retrieve and incorporate scientific concept relationships. The paper also collects approximately 100k papers from 10 major AI conferences to create such co-occurrence keywords. Finally, the paper released a review dataset based on real-world reviewer feedback.

**Strengths:**

1. The paper collects approximately 100k papers from 10 major AI conferences to create such co-occurrence keywords. Finally, the paper released a review dataset based on real-world reviewer feedback.
2. The paper includes an ablation study and a case study. The paper also includes several SOTA baselines. The paper conducts both human and automatic evaluation. The paper provides both quantitative and qualitative analysis.
3. The paper includes additional implementation details in the appendix.

**Weaknesses:**

1. Why use a co-occurrence concept graph instead of using scientific IE systems to extract keywords and their relationships? The framework seems to be purely based on prompting. The idea of using neighboring keywords has been proposed in ResearchAgent and SciMon. The paper might need to include a domain-specific LLM such as OLMO2.
2. Some details of the paper are also not clear. For the method section, the evolve part is really confusing. What is the exact algorithm? How to determine whether the keywords need to be removed? This part is also similar to the iterative idea refinement of ResearchAgent and SciMon. For the experiment section, what is the evaluation dataset? Are all of the 100k papers used for evaluation? What are the evaluation metrics used in Section 4.2? What is the background of human annotators? What is their inter-annotator agreement?
3. The paper fails to include a use of LLMs section. The paper also fails to provide code. The paper needs to include an ethics consideration.

**Questions:**

See weakness

---

### Official Review · Reviewer_DTQ4 · 2025-11-04

**Soundness:** 4
**Presentation:** 4
**Contribution:** 4
**Rating:** 6
**Confidence:** 5

**Summary:**

The paper introduces Deep Ideation, a novel LLM agent framework designed to generate high-quality research ideas by dynamically interacting with a vast scientific concept network. The framework constructs this network from approximately 100,000 papers, capturing contextual keyword relationships beyond simple co-occurrence. It employs an explore-expand-evolve iterative workflow, guided by an Idea Stack and a Critic Model trained on real-world reviewer feedback. Experimental results show a significant improvement in idea quality (10.67% over baselines) across multiple AI domains, with generated proposals exceeding the acceptance bar for top conferences.

**Strengths:**

1. The core innovation lies in integrating a comprehensive scientific concept network that incorporates contextual relationships between keywords, providing a richer, more grounded foundation for LLM ideation than previous statistical methods.
2. The explore-expand-evolve workflow, combined with an Idea Stack and keyword management modules, allows for a structured and continuous optimization of research ideas, mirroring the cognitive process of human researchers.
3. The introduction of a Critic Model trained on real-world reviewer feedback is a powerful mechanism to guide the search for novelty and feasibility, ensuring that the iterative process is not "directionless and blind".

**Weaknesses:**

1. The formal definition of the edge feature $F_{ij}$ (contextual relationship) is high-level ($g(\cdot)$ aggregating relation)9999999. The paper lacks detail on how this complex contextual information is practically represented, quantified, and distilled into a format that the LLM agent can robustly query and leverage for "Relation Analysis" beyond just raw text snippets.
2. The Critic Model is fine-tuned on a relatively small dataset (4278 examples). Its ability to consistently and accurately evaluate novelty and feasibility across diverse and potentially unrelated AI domains (the four domains tested) remains a key concern regarding generalization.

**Questions:**

1. The $F_{ij}$ feature captures the contextual relationship between co-occurring keywords12. Could the authors provide a case study or ablation study that specifically compares the performance of the ideation agent when using this complex $F_{ij}$ feature (contextual summary) versus a simpler edge feature (e.g., just the co-occurrence frequency or a semantic embedding of the connecting text)?
2. Given the iterative nature and multiple LLM calls per iteration (Relation Analysis, Keyword Selection, Idea Formulation, Critic Model), what is the amortized cost (e.g., total tokens/API cost) and time required to generate a single idea that achieves the "acceptance level" benchmark, compared to the one-shot or less complex iterative baselines?

---

### Official Review · Reviewer_gb37 · 2025-11-04

**Soundness:** 2
**Presentation:** 3
**Contribution:** 2
**Rating:** 2
**Confidence:** 4

**Summary:**

The paper proposes Deep Ideation, a framework integrating LLM agents with a scientific concept network for automated research idea generation. It constructs a network from 100k papers for keyword co-occurrences and contextual relations. The ideation loop follows an explore–expand–evolve process, with a Critic Model trained on peer reviews on Novelty & Feasibility (1-5 scale) to guide refinement.
Experiments across 4 AI domains against baselines SciMON, ResearchAgent, MOOSE-Chem, etc. Reported 8-10% improvements in LLM/human scores.

**Strengths:**

- The paper address an intereateing and important area of the LLM-driven research-idea generation
- Large corpus (100K papers) for the concept network, with modular design of explore / expand / evolve with critic feedback loop
- Clear visual pipeline for fig 1-2, easy to follow with good presence.
- Consistent experiment reporting across 4 domains with intent and structured scoring rubrics. The evaluation model reported is comprehensive.
- Ablations without critic show the contribution of the critic model

**Weaknesses:**

- Similar concept networks and iterative LLM ideation already exist. The method-wise novelty is somehow limited to incremental algorithmic refinements rather than conceptual advances.
- The critic is trained on LLM-generated paper–review pairs rather than real human reviews. Although an ablation (with vs. without critic) is included, the small (3–5%) improvement is measured only by LLM judges without correlation or reliability analysis. Also how to prevent the data contamination and model checkpoint date is not clear in the paper.
- The dataset statistics not reported.
- The paper scores ideas on novelty and feasibility, both defined through self-constructed rubrics without references to prior peer-review standards. These definitions are generic and may conflate novelty with simple graph distance. An Effectiveness —how meaningful, insightful, or actionable the generated ideas are— should also be inclused accroding to Si et al. 2024 (Can LLM generate novel research ideas?).
- Reported improvements are numerically minor and may fall within variance. The human evaluation details is very unclear (only mentions "meet the expectations of domain expert" is not enough, lacks details on blinding, recuiting, expertise, and statistical significance etc. ), making the evidence of real improvement very unconvincing. The paper does not report which dataset was used for the human study or how it aligns with the system evaluation set.
- The core graph is built from 100K papers, but the value of this corpus for actual idea generation is unclear. The model relies only on keyword co-occurrences and short LLM-generated contextual relations—mechanistically simple and lacking deeper conceptual or causal understanding. The aggregation function g(⋅) is undefined (I might miss it), and no analysis show the graph contributes meaningfully to novelty or feasibility.
- Comparisons are partly unfair (e.g., MOOSE-Chem is in a different domain). Other relevant baselines are missing—such as multi-agent simulation methods (“Many Heads Are Better Than One”), graph simulation systems (ResearchTown), and method-focused ideation such as IdeaSynth/WebWeaver/LDC.

**Questions:**

- What's the distribution of the dataset? this is very important to whether the graph construction make sense but not reported in the paper.
- How is relation(vi, vj, p) computed? Manual LLM parsing or embedding similarity?
- Is the critic model fine-tuned with actual numerical review scores or textual rationales only?
- How do you ensure generated ideas are not just rephrasings of seen papers but after the model checkpoints

---

### Note · Authors · 2025-11-27

I have read and agree with the venue's withdrawal policy on behalf of myself and my co-authors.